# Constructing molecular bridge for high-efficiency and stable perovskite solar cells based on P3HT

Dongdong Xu[1], Zhiming Gong[1], Yue Jiang [1]✉, Yancong Feng [2]✉, Zhen Wang[1], Xingsen Gao [1], Xubing Lu [1], Guofu Zhou[2], Jun-Ming Liu [3] & Jinwei Gao [1]✉

Poly (3-hexylthiophene) (P3HT) is one of the most attractive hole transport materials (HTMs) for the pursuit of stable, low-cost, and high-efficiency perovskite solar cells (PSCs). However, the poor contact and the severe recombination at P3HT/perovskite interface lead to a low power conversion efficiency (PCE). Thus, we construct a molecular bridge, 2-((7-(4-(bis(4-methoxyphenyl)amino)phenyl)−10-(2-(2-ethoxyethoxy)ethyl)−10H-phenoxazin-3-yl)methylene)malononitrile (MDN), whose malononitrile group can anchor the perovskite surface while the triphenylamine group can form π−π stacking with P3HT, to form a charge transport channel. In addition, MDN is also found effectively passivate the defects and reduce the recombination to a large extent. Finally, a PCE of 22.87% has been achieved with MDN-doped P3HT (M-P3HT) as HTM, much higher than the efficiency of PSCs with pristine P3HT. Furthermore, MDN gives the un-encapsulated device enhanced long-term stability that 92% of its initial efficiency maintain even after two months of aging at 75% relative humidity (RH) follow by one month of aging at 85% RH in the atmosphere, and the PCE does not change after operating at the maximum power point (MPP) under 1 sun illumination (~45 °C in N$_2$) over 500 hours.

Organic–inorganic hybrid perovskite solar cells (PSCs) with power conversion efficiency (PCE) over 25%[1-4] is widely considered as one of the most promising photovoltaic technologies for its low-cost manufacturing process and appealing photovoltaic performances. In the typical n–i–p type PSCs, hole transport materials (HTMs) are necessary that they can extract and transport holes to enhance the efficiency, and simultaneously block the moisture ingestion so as to be the last barrier of perovskite degradation. The doped 2,2',7,7'-tetrakis[N,N-di(4methoxylphenyl)amino]-9,9'-spirobifluorene (Spiro-OMeTAD) with appropriate hole mobility and energetic level alignment is the most commonly used HTM[5-8]. However, the achievement of high efficiency is always accompanied by substantial instability arising from the hygroscopicity and fluidity of dopants that can lead to perovskite

corrosion and dopant migration[9-13], which is detrimental to the long-term stability and commercialization of PSCs[14-18].

The development of dopant-free HTMs is thus viewed as an important solution. Organic conjugated materials with sophisticated structures thus have been designed and synthesized as dopant-free HTMs for efficient and stable PSCs. Those include small molecular systems, such as D–A–π–A–D-type DTP-C6Th[19], 1,10-phenanthroline (YZ22)[20], DTB-FL, Ni phthalocyanine (NiPc)[21], etc, and polymeric systems, such as 2DP-TDB[22], phenanthrocarbazole 6 (PC6)[23], Mes-TABT[24], etc. Interestingly, the common characteristic of all these materials is the claimed PCE, as high as over 21%, but none of them can really substitute the doped Spiro-OMeTAD, even on a lab scale. We conjecture this dilemma is caused by their delicate molecular structures,

[1]Institute for Advanced Materials and Guangdong Provincial Key Laboratory of Optical Information Materials and Technology, South China Academy of Advanced Optoelectronics, South China Normal University, Guangzhou 510006, China. [2]Institute of Electronic Paper Displays, South China Academy of Advanced Optoelectronics, South China Normal University, Guangzhou 510006, China. [3]Laboratory of Solid State Microstructures, Nanjing University, Nanjing 210093, China. ✉e-mail: yuejiang@m.scnu.edu.cn; fengyancong@m.scnu.edu.cn; gaojinwei@m.scnu.edu.cn

leading to the cumbersome synthesis and high cost, and their poor reproducibility on performances in PSCs due to the identical treatment. In addition, Zhang et al. reported a high PCE, ca. 20%, based on the undoped Spiro-OMeTAD through solvent-annealing assisted thermal evaporation[25]. However, this complex evaporation technique is not applicable for most laboratories and the reproducibility still needs verification.

In the meanwhile, utilizing the existing p-type semiconductors with high hole mobility and mature fabrication process, for example, poly (3-hexylthiophene) (P3HT), as HTMs is another significant attempt. Nevertheless, P3HT adopts the "edge-on" stacking arrangement, that is the alkyl side chains directly contact perovskite film, which was found presenting an electronically poor contact at perovskite/P3HT interface, which can aggravate the non-radiative recombination loss of PSCs[26]. Thus, the pristine P3HT as HTMs can only achieve a PCE as low as 16%[27,28].

In order to solve this problem, interfacial engineering is widely applied. For example, implementing BTCIC-4Cl[29] or copper(I) thiocyanate (CuSCN)[30] at perovskite/P3HT interface to passivate the surface defects of perovskite can give a reported efficiency of ~16% for CsPbI$_2$Br-based devices. Seo et al. introduced a layer of n-hexyl trimethyl ammonium bromide (HTAB) to modulate the packing of P3HT and obtained a PCE over 22% for (FAPbI$_3$)$_{0.95}$(MAPbBr$_3$)$_{0.05}$-based device[26]. And Sun et al. further doped P3HT with gallium(III) acetylacetonate (Ga(acac)$_3$) to reduce the interfacial recombination loss and demonstrated an enhanced PCE over 24%[31]. While, regarding the hole transport process, Hu group had improved the hole mobility by directly changing the "edge-on" packing of P3HT into "face-on" packing via SMe-TATPyr doping engineering[32]. Clearly, researchers have taken various strategies, passivation of perovskite defects or

modification of the P3HT hole mobility, to ameliorate the performance of P3HT in PSCs, whereas the poor contact issue at the perovskite/P3HT interface still has not been stressed.

Herein, we have introduced a molecular bridge (MDN) to electronically link perovskite films with P3HT. In this MDN bridge, the malononitrile group anchors perovskite and passivates its surface defects, while the triphenylamine (TPA) group is to form π–π stacking with the segments of P3HT. Finally, in PSCs, a PCE of 22.87% has been achieved with an enhanced open-circuit voltage ($V_{OC}$), from $0.88 \pm 0.03$ to $1.15 \pm 0.02$, and fill factor (*FF*), from $52.91\% \pm 5.14\%$ to $75.02\% \pm 3.09\%$. Moreover, the device has remained more than 90% of its initial efficiency when aging at 75% and then 85% RH (25 °C) over 2100 h.

## Results

### Molecular stacking structures

The target small molecules, MDN and RDN, are shown in Fig. 1a, and the detailed synthetic route was given in Supplementary Fig. 1. Among these, RDN was designed because of its similar molecular polarity with MDN to exclude the effect of the molecular polarity of MDN on its performance in PSCs. Their optical and electrochemical properties were then examined by UV–vis absorption spectrometer and cyclic voltammetry, where MDN and RDN were determined to have the appropriate energetic levels, ca. the highest occupied molecular orbital of −5.31 and −5.26 eV and the lowest unoccupied molecular orbital of −3.28 and −3.24 eV, matching well with perovskite (Supplementary Figs. 2 and 3 and Table 1).

Further, we added MDN (4 mg) and RDN (4 mg) in P3HT (10 mg ml$^{-1}$), separately, denoted as M-P3HT and R-P3HT, and compared their optical and energetic structures. The packing structures

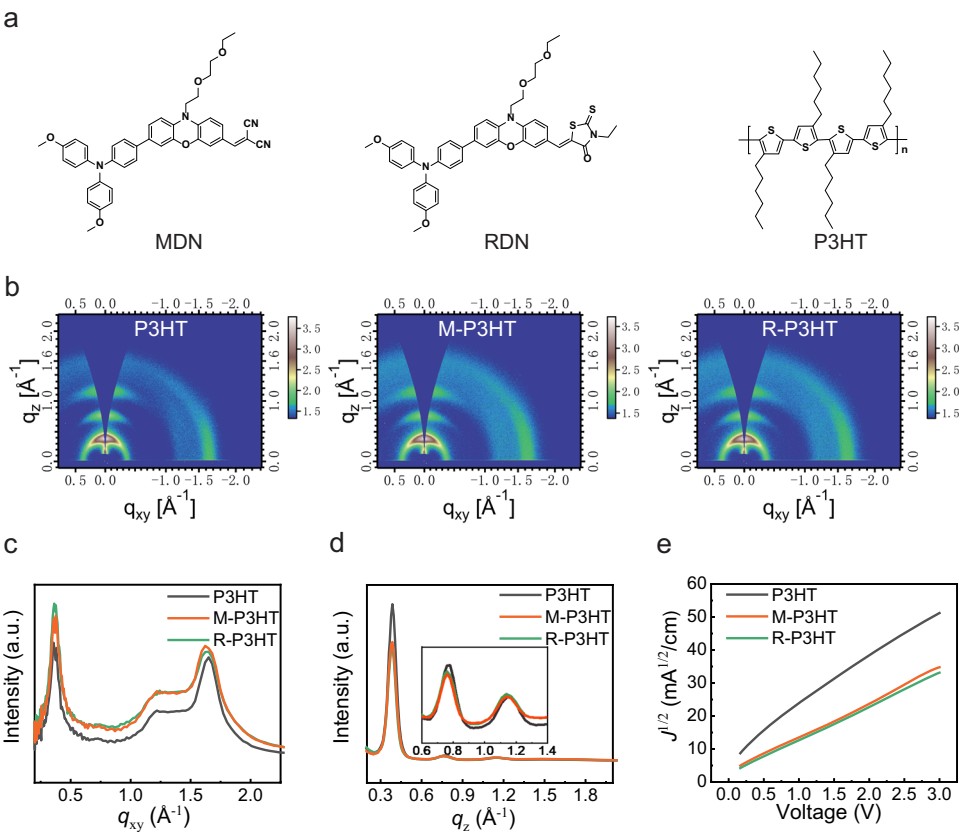

**Fig. 1 | Molecular packing characterization of P3HT, M-P3HT, and R-P3HT.** **a** Molecular structure of MDN, RDN, and P3HT. **b** 2D GIWAXS patterns. **c** 1D intensity profiles of relevant films along the in-plane and **d** out-of-plane directions. **e** Hole mobility measurement characteristics by the space charge limited current (SCLC) method.

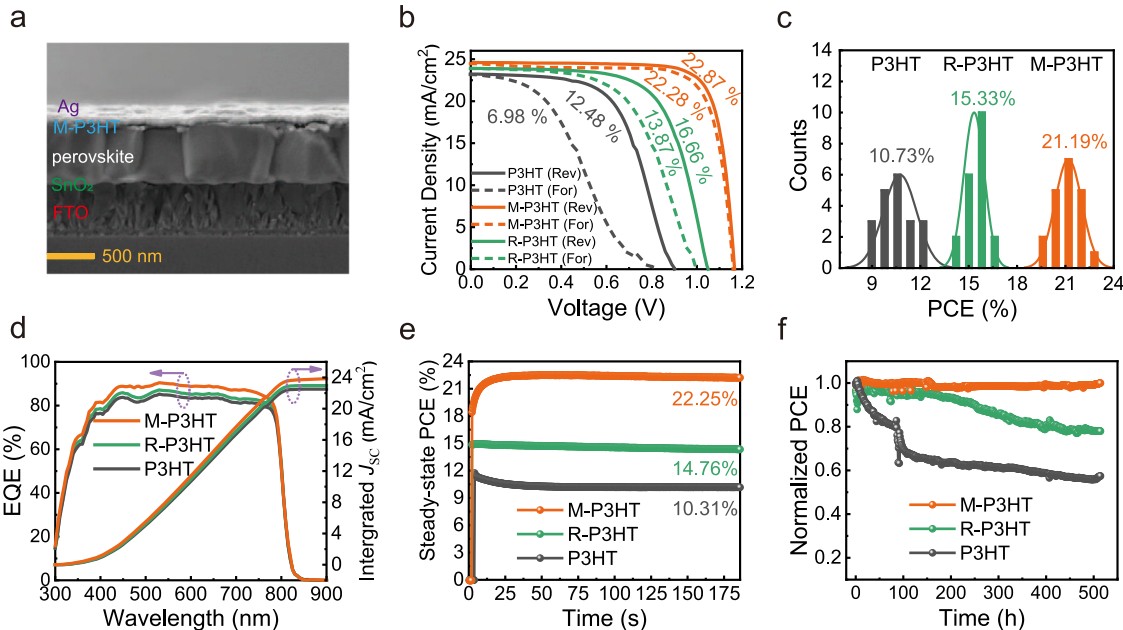

**Fig. 2 | PSCs performances. a** Cross-sectional SEM image of PSC with M-P3HT. **b** J−V curves of the champion devices based on P3HT, M-P3HT, and R-P3HT, respectively. **c** PCE histogram was obtained from 20 PSCs. **d** EQE spectra and integrated photocurrent curves of the device with P3HT, M-P3HT, and R-P3HT. **e** The steady-state PCE. **f** The device's stability at maximum power point (MPP) under full solar illumination (AM 1.5 G, 100 mW cm$^{-2}$) in the glovebox.

of M-P3HT and R-P3HT on silicon wafers were compared with bare P3HT via grazing-incidence wide-angle X-ray scattering. From Fig. 1b, c, the obvious in-plane (010) diffraction rings of P3HT, M-P3HT, and R-P3HT are located at $q_{xy} = 1.65$, 1.62, and 1.64 Å$^{-1}$, with the corresponding π−π stacking distances calculated to be 3.81, 3.88 and 3.83 Å, respectively. And the out-of-plane (100), (200), and (300) diffraction rings of P3HT, $q_z = 0.387$, 0.771, and 1.166 Å$^{-1}$, are similar to those of M-P3HT and R-P3HT, both $q_z = 0.387$, 0.761 and 1.138 Å$^{-1}$ (Fig. 1d). In addition, P3HT, M-P3HT and R-P3HT deposited on perovskite (Cs$_{0.05}$FA$_{0.85}$MA$_{0.10}$Pb(Br$_{0.03}$I$_{0.97}$)$_3$) have given the same stacking arrangement as illustrated in Supplementary Fig. 4. Thus, MDN and RDN barely changed the "edge-on" packing structure of P3HT.

Simultaneously, their optical and electronic properties were compared. As shown in Supplementary Figs. 5–7, through UV−vis and ultraviolet photoelectron spectroscopy measurements, no significant difference can be distinguished between P3HT, M-P3HT, and R-P3HT. Similarly, the hole mobility was measured by the space charge limited current method based on the hole-only device structure of ITO/PEDOT:PSS/HTM/Ag. And the hole mobilities of $3.03 \times 10^{-3}$ cm$^2$ V$^{-1}$ s$^{-1}$ for P3HT and ~$2.88 \times 10^{-3}$ cm$^2$ V$^{-1}$ s$^{-1}$ for M-P3HT or R-P3HT were obtained (Fig. 1e). This slightly smaller hole mobility is consistent with the minute expansion of the π−π stacking distance when MDN and RDN were adopted. Moreover, similar conductivities were observed for P3HT, M-P3HT, and R-P3HT films (Supplementary Fig. 8). Therefore, the addition of MDN and RDN has a negligible influence on the physical properties of P3HT.

## Photovoltaic performances

The photovoltaic performance of M-P3HT and R-P3HT HTMs in PSCs has been examined by fabricating n−i−p type devices with the configuration of FTO/SnO$_2$/Cs$_{0.05}$FA$_{0.85}$MA$_{0.10}$Pb(Br$_{0.03}$I$_{0.97}$)$_3$/HTM/Ag, as shown in the cross-sectional scanning electron microscopy (SEM) image (Fig. 2a and Supplementary Fig. 9). The thickness of each layer was estimated to be 560, 20, 610, 30, and 100 nm, respectively. The top view SEM images in Supplementary Fig. 10 have clearly demonstrated a smooth and even surface of M-P3HT and R-P3HT as P3HT.

The J−V curves of the champion devices have been plotted in Fig. 2b, that the PCEs of devices based on P3HT, M-P3HT and R-P3HT HTMs are 12.48% ($V_{OC} = 0.92$ V, $J_{SC} = 23.21$ mA cm$^{-2}$, $FF = 58.73\%$), 22.87% ($V_{OC} = 1.16$ V, $J_{SC} = 24.58$ mA cm$^{-2}$, $FF = 80.17\%$) and 16.66% ($V_{OC} = 1.05$ V, $J_{SC} = 23.90$ mA cm$^{-2}$, $FF = 66.54\%$), respectively. Obviously, all the parameters, including $V_{OC}$, $J_{SC}$, and $FF$, have been enhanced in PSCs with M-P3HT, which can be highly related to the efficient charge dynamic process throughout the devices. Additionally, a significantly mitigated hysteresis was observed when adopting M-P3HT as HTM, with its hysteresis index as low as 2.6%, compared with those of P3HT and R-P3HT, ca. 78.8% and 20.1% (Supplementary Table 2). This can be explained with the constructed molecular bridge for efficient carrier transportation as well as the defect passivation[33–35]. The external quantum efficiency spectra along with the integrated $J_{SC}$, ca. 22.6, 23.9, 22.9 mA cm$^{-2}$ for the device with P3HT, M-P3HT, and P-P3HT, further confirm the better performance of M-P3HT (Fig. 2d).

By analyzing the statistics of 20 independent devices prepared at the same condition, the reproducibility of PSCs with P3HT, M-P3HT, and R-P3HT has been demonstrated. Figure 2c presents the histogram of the PCEs, where the average efficiency of P3HT is only 10.73%, much lower than that of R-P3HT (15.33%) and M-P3HT (21.19%). The average $V_{OC}$ and $FF$ show the same trend (Supplementary Figs. 11 and 12 and Table 3), agreeing well with the J−V curves in Fig. 2b. Figure 2e shows the steady power output of PSCs with various HTMs at their maximum power point (MPP) bias over 180 s. The devices with P3HT, M-P3HT, and R-P3HT have achieved a stable power output of 10.31%, 22.25%, and 14.76%, respectively. The devices were then placed in a nitrogen atmosphere and subjected to a light stability test at MPP. As shown in Fig. 2f, the device based on M-P3HT shows the nearly same efficiency as its starting state even after more than 500 h of light soaking, while the other two kinds of devices present severe efficiency degradation.

## Molecular bridge of MDN

Based on the above results, it is obvious that MDN plays a significant role in PSCs with P3HT HTM. Hence, we utilized X-ray photoelectron spectroscopy (XPS) to detect the surface of perovskite. When capped with P3HT or R-P3HT, the binding energies (BEs) of Pb 4$f$ and I 3$d$

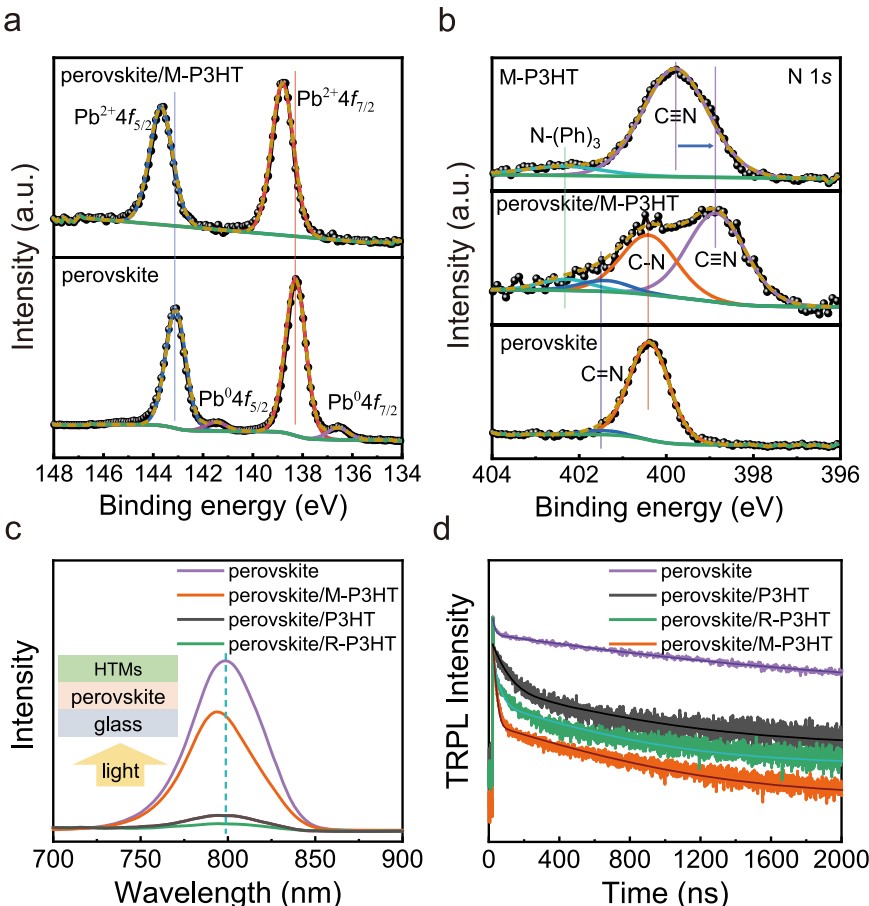

**Fig. 3 | Molecular bridge characterization of M-P3HT.** High-resolution XPS spectra of **a** Pb 4*f* spectra, and **b** N 1*s* spectra. **c** Steady-state PL spectra, and **d** TRPL spectra.

barely changed compared with pristine perovskite film (Supplementary Fig. 13), which is different from the case with M-P3HT. In Fig. 3a and Supplementary Fig. 14, the BEs of Pb 4*f* and I 3*d* shift to higher values with 560 and 330 meV indicating a prominent change in the electronic environment of the perovskite surface[36]. Simultaneously, the metallic Pb[0] with BEs at 141.51 and 136.59 eV that arises from the decomposition of PbI$_2$ caused by light and X-ray irradiation has been effectively inhibited by M-P3HT (Fig. 3a)[37].

In addition, from the XPS spectra of perovskite, perovskite/P3HT, and perovskite/M-P3HT in Fig. 3b, the BEs of N 1*s* peaks based on perovskite/M-P3HT is different from others. By peak splitting, perovskite/M-P3HT shows the characteristic peaks of C=N at 401.53 eV and C−N at 400.36 eV from perovskite films, and the peaks of N−(Ph)$_3$ at 402.36 eV and C≡N at 398.79 eV ascribing to M-P3HT[38,39]. More interestingly, this BE of the C≡N bond is smaller than that of pristine M-P3HT (399.77 eV). Combining with a stronger BE of Pb 4*f* in perovskite/M-P3HT (Fig. 3a), it is concluded that an electrostatic coupling between the N atoms with high electron cloud density in the malononitrile group of MDN and the uncoordinated Pb exists, and therefore, building a charge transportation path between perovskite and M-P3HT.

We have performed the steady-state photoluminescence (PL) and time-resolved PL (TRPL) to investigate the charge transfer kinetics at perovskite/HTM interface. To avoid the light absorption by P3HT and consist with the *J*−*V* characterization, the illumination was applied from the perovskite side. Figure 3c shows the PL spectra, where different from the low intensity of device with P3HT and R-P3HT due to their charge extraction effect, M-P3HT exhibits a strong PL intensity along with a significant blue-shift, from 799 to 792 nm, which indicates accompanied with a charge extraction process, the trap state density

at the perovskite/M-P3HT interface has also decreased[40,41]. In Fig. 3d, the deconvolution fitting from the result of TRPL decay (Supplementary Table 4), and the average PL lifetime of perovskite, perovskite with P3HT, R-P3HT, and M-P3HT HTMs were determined to be 1071.71, 464.75, 397.38, and 302.377 ns, respectively, indicating the faster carrier transfer from perovskite to HTM when MDN was adopted. In particular, the shorter PL lifetime (τ1) correlates to the non-radiative recombination[41–43], suggesting the mitigated non-radiative recombination at perovskite/M-P3HT interface, consistent with the steady-state PL results. This is direct evidence of the formation of molecular bridge paths by MDN.

This charge transportation path is also verified by inserting a thin (to ensure the charge tunneling effect) layer of inert polystyrene (PS) between perovskite and M-P3HT in order to isolate their direct contact. As shown in Supplementary Figs. 15 and 16 and Tables 5 and 6, after PS modification, the device efficiency of PSCs with P3HT HTM increases, and the efficiency keeps constant with R-P3HT HTM, whereas the photovoltaic performance of PSCs with M-P3HT declines significantly, that the average efficiency drops from 21.19 to 18.35%. Therefore, these results confirm that this electronic coupling built by MDN and perovskite has played a significant role in the photovoltaic performance of PSCs (Supplementary Fig. 20a).

Further, MDN and RDN are inserted as an interface layer between perovskite and P3HT to build the PSCs. It is found that RDN only improves interfacial contact with PCE increases from 10.73 to 13.45%, while for MDN, the PCE is 16.30%, much lower than that of the PSC with 22.87% of M-P3HT (Supplementary Figs. 17 and 18 and Tables 7 and 8). We then explored the MDN and MDN/P3HT stacking patterns, as shown in Supplementary Fig. 19, where first the MDN is stacked out of order, while the P3HT appears to change from edge-on to disorder

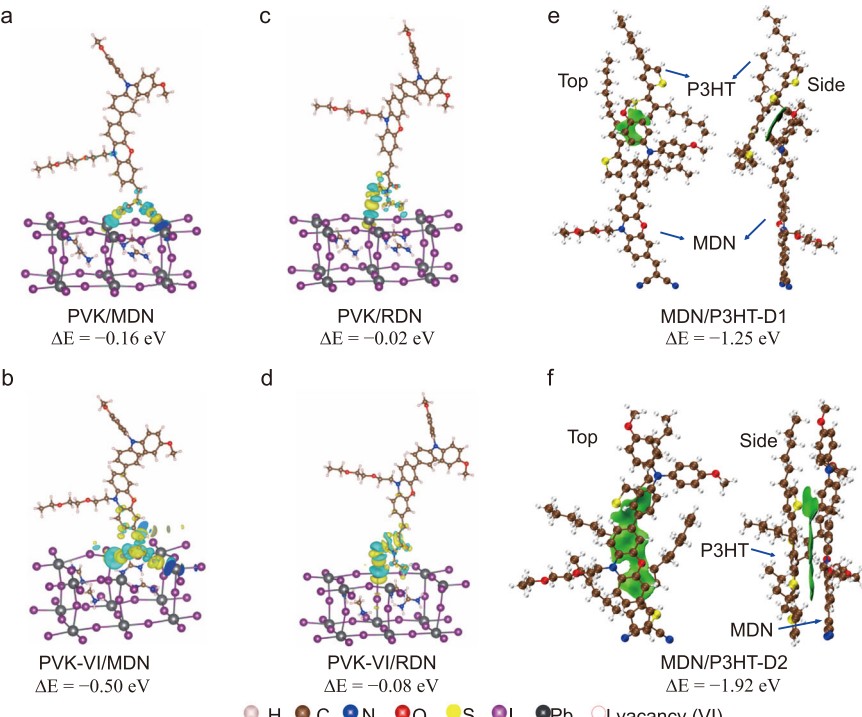

**Fig. 4 | Snapshots of the molecular bridge.** Charge density differences of **a** PVK/MDN, **b** PVK-VI/MDN, **c** PVK/RDN, and **d** PVK-VI/RDN. PVK-VI denotes the perovskite with I vacancy, and ΔE expresses the binding energy. Local perovskite is displayed for clarity. The yellow and cyan regions represent electron accumulation and depletion, respectively. Independent gradient model based on Hirshfeld partition (IGMH) of MDN/P3HT dimer complexes with different geometries **e** and **f**.

after spin-coating on the MDN. It is clear that as an interfacial layer, MDN fails to form a charge transportation channel, which is possibly due to its disordered stacking with P3HT, as shown in Supplementary Fig. 20b.

DFT calculations were adopted to further investigate the interactions between MDN/RDN, P3HT, and perovskite. Supplementary Figs. 21–24 show the optimized structures of perovskite–molecule (MDN/RDN) dimer complexes, where the electron-withdrawing groups, di-cyanovinyl and 3-ethylrhodanine, of MDN or RDN anchor the PbI-terminated perovskite surface, respectively. Figure 4a–d presents the charge density differences between perovskite (with and without I vacancies, VI) and MDN/RDN, named as PVK-VI/MDN, PVK/MDN, PVK-VI/RDN, and PVK/RDN, respectively[44]. It can be seen that the electron-withdrawing groups of MDN/RDN attract the electron from Pb ion in perovskite, while the degree of charge transfer in MDN- dimer complexes is greater than that in RDN- one. Meanwhile, the Pb ion in perovskite with I vacancies has more electron depletion than that without I vacancies. And the lowest BE (ΔE) can be obtained in PVK-VI/ MDN, indicating their strongest interaction to let MDN anchor on the perovskite surface.

While on the other side, the interaction between MDN and P3HT was analyzed by DFT calculations. From the localized orbital locator (LOL) in Supplementary Fig. 25, MDN has shown a thorough π-conjugated structure. And in Fig. 4e, f, π–π stacking of MDN and P3HT can be observed via the analysis of the independent gradient model based on Hirshfeld partition (IGMH)[45]. In particular, we find that the triphenylamine group of MDN has a greater degree of π–π stacking and stronger interaction with P3HT when comparing the geometries in Fig. 4e, f.

To conclude, despite RDN and MDN having similar polarity and optical and electronic properties, MDN can not only anchor the surface of perovskite but also π–π stack with P3HT, forming a charge transportation channel as a molecular bridge. This explains the fact that when MDN was merely deposited as the interfacial layer between

perovskite and P3HT, in the PSCs, its PCE is not as high as that based on M-P3HT due to its poor π–π stacking with P3HT.

Further, we have checked D–D–A molecular structure of MDN by design and synthesis of D–A and π–A molecules, PDN, and TDN (Supplementary Fig. 26). After utilizing PDN and TDN in PSCs with the same procedure as MDN, it is found that they could not get a similar effect as MDN (Supplementary Figs. 27 and 28 and Tables 9 and 10), suggesting the uniqueness of the MDN.

## Defect passivation and stability

On account of the interaction between MDN and perovskite, the defect density of perovskite was estimated with hole-only devices by the space-charge limited current method. The defect densities ($N_t$) can be calculated from the trap-filled limit voltage ($V_{TFL}$) in Fig. 5a, corresponding to $1.39 \times 10^{16}$ cm$^{-3}$, $9.21 \times 10^{14}$ cm$^{-3}$, and $6.35 \times 10^{15}$ cm$^{-3}$ for the devices with P3HT, M-P3HT, and R-P3HT, respectively. The obtained ultra-low trap density of a device with M-P3HT, only 1/15 of that with P3HT, explains the minimal non-radiative recombination[46]. Besides, their hole mobility was calculated from the curves in the Child region (Supplementary Fig. 29), ca. $9.86 \times 10^{-6}$ cm$^2$ V$^{-1}$ s$^{-1}$, $4.10 \times 10^{-4}$ cm$^2$ V$^{-1}$ s$^{-1}$, $1.20 \times 10^{-5}$ cm$^2$ V$^{-1}$ s$^{-1}$, respectively, further confirming the MDN molecular bridge.

Electrical impedance spectroscopy has been applied to assess the carrier transport dynamics. The Nyquist plot in Fig. 5b shows that M-P3HT HTM gave PSCs a smaller transfer resistance ($R_{tr}$) and a larger recombination resistance ($R_{rec}$). Besides, the capacitance–frequency plots (Fig. 5c) present 2 orders of magnitude decrease of capacitance at low frequency for devices with M-P3HT or R-P3HT when compared with conventional PSC based on P3HT HTM, owing to the fact that MDN brings about effective defect passivation along with the formation of charge transfer channels, resulting in reducing the charge accumulation at perovskite/HTM interface[25,47].

The voltage dependence on light intensity was plotted in Fig. 5d to investigate the charge recombination behavior in the PSCs. It is clear

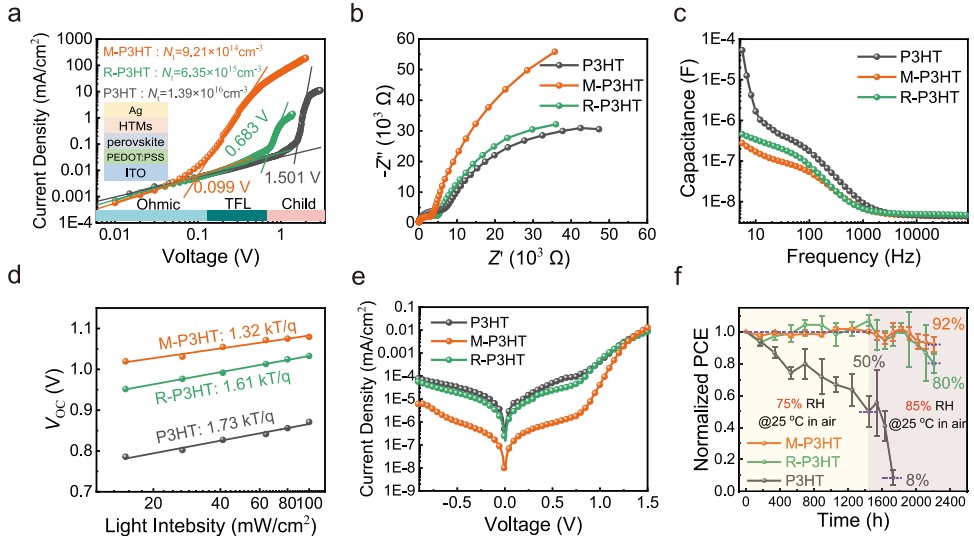

**Fig. 5 | Passivation of perovskite surface defect and stability of devices. a** $I$–$V$ curves of the hole-only devices for quantitative measurement of the trap densities. **b** Nyquist plots of PSCs under the dark condition at 0.8 V bias. **c** Capacitance–frequency plots of PSC devices employing different HTM. **d** The dependence of $V_{OC}$ on the light intensity of relevant PSCs. **e** Dark $J$–$V$ curves of the devices. **f** Stability of the devices in high humidity gradient environment without encapsulation. The error bars represent the standard deviations from ten samples for each condition.

that M-P3HT-based devices exhibit higher $V_{OC}$ values than the devices based on P3HT and R-P3HT, indicative of less recombination during the photovoltaic process[48,49]. Moreover, the smaller linear slope of PSC with M-P3HT (1.32 kT $q^{-1}$) than that with R-P3HT (1.61 kT $q^{-1}$) and P3HT (1.73 kT $q^{-1}$) further verified this argument.

While the leakage current density obtained from $J$–$V$ curves measured in dark is much lower in devices with M-P3HT, its saturation current density ($J_0 = 9.7 \times 10^{-9}$ mA cm$^{-2}$) is also 2 orders of magnitude lower compared with PSCs with P3HT and R-P3HT ($J_0 = 3.6 \times 10^{-7}$ mA cm$^{-2}$, $J_0 = 1.5 \times 10^{-7}$ mA cm$^{-2}$), as shown in Fig. 5e. And the current density of M-P3HT device increases significantly in the high voltage region (0.8–1.5 V), indicating that the perovskite/M-P3HT interface gives a better charge transportation[50,51].

At last, to investigate the influence of HTM on a device's long-term stability in a moisture atmosphere, we monitored the PCE of PSCs with various HTMs without any encapsulation at a high relative humidity gradient (75% RH for 60 days and then 85% RH for 30 days) at 25 °C in the atmosphere. As shown in Fig. 5f, the efficiency of M-P3HT and R-P3HT-based devices hardly changed during the first 2 months, which is in sharp contrast with the situation for P3HT-based devices, where only 50% of the initial efficiency was maintained. More importantly, further increasing the relative humidity to 85%, with the protection of M-P3HT and R-P3HT, their PSCs still exhibited ~92% and ~80% of their initial efficiency after another 1-month aging, showing an effective moisture blockage of M-P3HT. We then examined the surface morphology of the aged devices and found that both the P3HT and R-P3HT-based devices showed some holes, which is possibly coming from the degradation of perovskite layers (Supplementary Fig. 30). By comparison, the surface of M-P3HT is still continuous and homogeneous, which means that M-P3HT could favor to inhibit the degradation of perovskite films. This enhanced stability is considered to come from the high hydrophobicity of M-P3HT (Supplementary Fig. 31), the tighter molecular structure, and the reduction of perovskite surface defects which acts as the starting sites for perovskite decomposition.

## Discussion
In summary, we have constructed a molecular bridge, MDN, to improve the poor contact between perovskite and P3HT, where the malononitrile group of MDN anchors the surface of perovskite and the triphenylamine group forms a tight π–π stacking with the conjugated polythiophene segment in P3HT. And thanks to this charge transportation channel, the fabricated PSCs have achieved a PCE as high as 22.87% with negligible hysteresis. Moreover, the unencapsulated PSCs have shown long-term stability at high relative humidity in the atmosphere, that more than 90% of the efficiency has been maintained over 3 months of aging, as well as light stability over 500 h at MPP. These distinguished results clearly show a bright future for the industrialization of low-cost, stable, and efficient PSCs by utilizing commercial P3HT-based HTMs.

## Methods
### Materials
The Cs$_{0.05}$FA$_{0.85}$MA$_{0.10}$Pb(Br$_{0.03}$I$_{0.97}$)$_3$ solution was formulated by adding 742.2 mg PbI2 (99.999%), 224.4 mg FAI (≥99.5%), 16.2 mg MABr (≥99.5%), 20.3 mg MACl (≥99.5%) and 19.8 mg CsI (99.999%) in a mixture of 800 ul dimethyl formamide DMF (99.8%) and 200 ul dimethyl sulfoxide DMSO (99.9%), and then shaking at room temperature to dissolve them. The electronic-transport material (ETM) is SnO$_2$, which solution was obtained using SnCl$_2$·2H$_2$O (Alfa Aesar)[52]. The HTM solution was constructed addition of 4 mg MDN and 10 mg P3HT to 1 ml chlorobenzene (CB), and then stirred at 60 °C to improve solubility. The remaining exploratory concentrations were 2 mg MDN (10 mg P3HT) and 6 mg MDN (10 mg P3HT), and RDN-P3HT was prepared by 4 mg RDN and 10 mg P3HT. We call P3HT with MDN as M-P3HT, and P3HT with RDN as R-P3HT. And we call P3HT with PDN as P-P3HT, and P3HT with TDN as T-P3HT. 2-P-P3HT, 4-P-P3HT, and 6-P-P3HT represent 2 mg, 4 mg, and 6 mg of PDN added to 10 mg P3HT and then dissolved in 1 ml CB, respectively. 1-T-P3HT, 2-T-P3HT, and 3-T-P3HT represent 1 mg, 2 mg, and 3 mg of TDN added to 10 mg P3HT and then dissolved in 1 ml CB, respectively. In addition, the polystyrene (10 mg ml$^{-1}$) interface and MDN/RDN interface were spin-coated on the perovskite surface at 3000 rpm for 20 s.

### Perovskite solar cells fabrication
Firstly, the FTO (7 Ω sq$^{-1}$) was cleaned by detergent, deionized water, and isopropanol for ~15 min each under ultrasonic, and then blow-dried with nitrogen gas. The ETM was then spin-coated with SnO$_2$ in the atmosphere and annealed under 150 °C (30–60 min). And next, the perovskite precursor solution (1000 rpm, 10 s, and 5000 rpm, 30 s) was dropped on the SnO$_2$ layer, with 100 μl chlorobenzene as

antisolvent, and annealed at 150 °C for about 10 min to form a photovoltaic-active black phase of perovskite. Afterward, the HTMs were spin-coated (3000 rpm, 20 s) onto the perovskite layers. At last, the Ag film ~100 nm thick was thermally evaporated under a vacuum. A metal aperture mask with an area of 0.07 cm$^2$ was used for testing.

## Computational details

Density functional theory calculations for perovskite/molecule systems were performed by the Vienna ab initio simulation package. The projector-augmented wave methods was adopted, and the Perdew-Burke-Ernzerhof of the generalized gradient approximation was selected. Primitive cell (FA$_{0.875}$MA$_{0.125}$PbI$_3$) was adopted based on the result of our previous work[44]. The slab models consist of $(2 \times 2 \times 1)$ supercells with a vacuum of 20 Å in the [010] direction. The k-point mesh of $2 \times 1 \times 2$ and the plane wave energy cutoff of 400 eV were used to achieve energy and force convergence of 0.1 meV and 0.02 eV Å$^{-1}$, respectively. In our models, the electron-withdrawing group of bridging molecules, MDN and RDN molecules, anchor the PbI-terminated perovskite surface, respectively. Supplementary Fig. 32 shows the primitive cell of FA$_{0.875}$MA$_{0.125}$PbI$_3$. The ratio of FA to MA ion is 0.875:0.125, which is close to the experimental value (0.85:0.10). Supplementary Fig. 33 presents the slab model consisting of $(2 \times 2 \times 1)$ supercells with a vacuum of 20 Å in the [010] direction after structural optimization.

The calculations of molecular systems were performed using the Gaussian code. B3LYP exchange-correlation functional with Grimme's empirical dispersion correction, which is abbreviated as B3LYP-D3(BJ), and the 6-311g** basis set were selected. The analyses of the independent gradient model based on Hirshfeld partition (IGMH) and LOL were carried out by Multiwfn code[53]. The maps were rendered using the VMD code based on the files exported by Multiwfn.

## Reporting summary

Further information on research design is available in the Nature Portfolio Reporting Summary linked to this article.

## Data availability

All data generated in this study are provided in the article and Supplementary Information, and the raw data supporting this study are available from the Source Data file. Source data are provided with this paper.

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

## Acknowledgements

This work is supported by NSFC Funds (U1801256) (J.G.), Guangdong Basic and Applied Basic Research Foundation (2022A1515010264) (Y.J.), Science and Technology Projects in Guangzhou (202002030130 (Y.J.), 202201000008 (J.G.)), Guangdong Provincial Key Laboratory of Optical Information Materials and Technology (2017B030301007) (G.Z.). We also thank the support from the Guangdong Provincial Engineering Technology Research Center for Transparent Conductive Materials (J.G) and the support from Office of International Exchange & Cooperation in South China Normal University (Y.J.).

## Author contributions

D.X., Z.G., Y.J., and J.G. conducted the idea. Z.G. and D.X. designed and synthesized target molecules. D.X. performed the electro-optical characterizations and the experiments on solar cells. Y.F. performed the DFT calculation. Z.W. coordinated the research activity. X.G., X.L., G.Z., and J.L. contributed to the revision of the manuscript. The manuscript was written by D.X., Y.J., and J.G. Y.J and J.G. directed the work. All the authors approved the final version of the manuscript.

## Competing interests

The authors declare no competing interests.
