## [Peer Review File · Nature Communications]

Constructing Molecular Bridge for High-Efficiency and Stable Perovskite Solar Cells based on P3HTREVIEWER COMMENTS

Reviewer #1 (Remarks to the Author):

A manuscript entitled “Constructing Molecular Bridge for High-Efficiency and Stable Perovskite Solar Cells based on P3HT Hole Transport Material” reports an improved performance and stability by introducing MDN-blended P3HT as hole transport material in the device architecture. Though MDN usage as an additive in P3HT could be the first case from this manuscript, this strategy seems not to provide a breakthrough in terms of concept, economic prospect, and performance/stability. Furthermore, the experimental results shown in this manuscript remain questionable points. The points could be found below the list. Therefore, as considering Nature Communication criteria for publication, regretfully, this manuscript does not correspond to the criteria.

1) A sentence “P3HT adopts the “edge-in” stacking ~ the non-radiative recombination loss of PSCs.” could deliver inappropriate meaning. Although ‘edge-on’ orientation is derived from direct contact between alkyl side chain and perovskite film, the evidence is still scarce to demonstrate how the orientation correlates to the junction properties.

2) Figure 1 (GIWAXS & SCLC) shows almost identical crystalline properties of P3HT layers regardless of molecular additives and the estimated hole mobilities are in the marginal range as well. So, further description must be required to help to understand why the quite similar results are important for this work. If not, a part related to Figure 1 might make confusion for readers.

3) It seems that the XPS result of Pb element and the description are opposite as compared to a precedent (<https://doi.org/10.1038/s41467-019-13909-5>). The previous paper suggested a malononitrile group in 6TIC-4F leads to lower binding energy of Pb element after being coated on a perovskite surface.

4) A sentence “it is clear that as interfacial layer, ~ as shown in Fig. S18b.” should be revised because GIWAXS pattern of M-P3HT does not demonstrate more ordered stacking as compared to that of pristine P3HT.

Reviewer #2 (Remarks to the Author):

This work titled "Constructing Molecular Bridge for High-Efficiency and Stable Perovskite Solar Cells based on P3HT Hole Transport Material" By Xu et al. develops and demonstrates a molecular bridge, MDN, between perovskite film and P3HT HTM for the n-i-p type perovskite solar cells, aiming to

substitute the un-stable Spiro-OMeTAD HTM with a more stable and widely used polymeric semiconductor P3HT. The addition of this molecular bridge has shown several advantages, such as the significantly enhanced efficiency, from ~12% to >22%, improved stability at high humidity environment. The authors have also compared the MDN with RDN which has the similar molecular structure and polarity, thus claiming that the improvement of efficiency is highly related to the electronic transporting channel due to the designed molecular structure. The area of research is important that from the respect of stability of the efficient n-i-p type PSCs, the application of Spiro-OMeTAD poses severe degradation issue in a long-term period. This work has shown impressive efficiency and stability results, and a clear demonstration of the underlined mechanism, which can provide a guideline for design new efficient HTMs. I recommend acceptance this manuscript after the following minor comments are properly considered or addressed.

1. It is clear that the stability of the PSCs based on M-P3HT HTM is enhanced. However, the insertion of MDN is more like doping in P3HT at a specific location, ca. at the perovskite/P3HT interface, the morphology stability of M-P3HT will be an important cause to the overall degradation of the PSCs. The authors should give a more detailed investigation about the degradation process of PSCs based on M-P3HT, especially from the respect of its morphology stability.
2. As listed in the introduction, such as references 29-32, the authors have compared several solutions to improve the efficiency based on P3HT. Clearly, some work has obtained a good result. I would suggest the authors to demonstrate the difference in order to highlight the novelty of the MDN molecular bridge.
3. The authors have also designed a series of D-A and -A molecules to prove that the D-D-A structure of MDN is an important reason of the enhanced device performance. Nevertheless, the design clue of the MDN is not clear. To make this work more scientific, I would suggest the authors give a clear demonstration about the molecular structure.
4. In the PV performances part, it is interesting to notice that the hysteresis of the PSCs based on M-P3HT has been significantly mitigated. As far as I know, ion migration process could be an important reason, however, it is obvious that the adopted perovskite is the same. Therefore, what is the origin of this big difference between PSCs with P3HT, M-P3HT, and R-P3HT. The authors should give an explanation.
5. Despite the authors have shown a good result based on the MDN doped P3HT, the physical characterization of MDN is limited. Especially, the stability of MDN is of great importance, that its decomposition could possibly cause the degradation of perovskite films. Thus, the stability of the pure MDN or RDN should also been investigated, at least, the analysis of TGA to illustrate the decomposition temperature, should be conducted.
6. For all the NMR spectra, the reference is all missing.
7. In the supporting information, the method of CV characterization, "Ferrocene/ferrocenium (Fc/Fc+) was used as the reference", here I think the Ferrocene/ferrocenium (Fc/Fc+) was acting as a calibration or a reference electrode. Please clarify.

8. Figure S11, the caption "J-V characteristics of the best-performing." Please complete the sentence.

Reviewer #3 (Remarks to the Author):

The authors introduced a molecular bridge into the interface of the perovskite/P3HT for providing a hole extraction channel from perovskite surface to P3HT HTL, thereby improving the photovoltaic performance. As described in the manuscript, a poor contact at the interface of the perovskite/P3HT is a major reason for a low PCE of the perovskite solar cells using P3HT as HTL. The well-designed molecular bridge including dicyanovinyl and triphenylamine moiety was developed to successfully reduce the surface defects of the perovskite film and induce the strong interaction with P3HT without sacrificing the ordering of P3HT, which could deliver a high power-conversion efficiency of 22.87% and improved stability against humid air and light soaking. The concept, results and discussion given in this paper are clear and seem to be reasonable. Thus, I would recommend the publication of this work in Nature Comm. after the following issues are well-addressed.

- (1) The authors need to state more discussion on the kinetics of the hole extraction at the interface of the perovskite/P3HT by carrying out TRPL or TA studies.
- (2) In Figure S12 and Figure 3, it is found that MDN is more effective in passivating the perovskite surface than RDN. For clear comparison, the authors need to provide TCSPC measurement for two samples.
- (3) In page 10, "It is clear that as interfacial layer, MDN fails to form a charge transportation channel, which is possibly due to its disordered stacking with P3HT, as shown in Fig. S18b." is mentioned. Did the authors compare the ordering of P3HT on MDN/perovskite with the ordering of M-P3HT on the perovskite film?
- (4) It is hard for the readers to identify the labelling and colors of MDN/P3HT complex in Figure 4e and 4f. It would be helpful if extended picture with a higher resolution is larger. In addition, top view SEM images of Figure S9 with higher resolution need to be given.
- (5) In Figure 2a and Figure S8, the thickness of P3HT is found to be ~ 30 nm. It is also hard to distinguish P3HT layer from three layers of perovskite, P3HT and Ag. Is a thin (~ 30 nm) layer of P3HT enough to cover the rough surface of the perovskite film? The authors need to make this clearer.
- (6) In Figure 5e, dark current densities of P3HT and R-P3HT devices are much higher than that of M-P3HT device. What is the main reason for this difference? Is such a trend consistent with SCLC results from Figure S27?

Response to the reviewers' comments for the paper entitled
**“Constructing Molecular Bridge for High-Efficiency and Stable Perovskite Solar Cells
based on P3HT Hole Transport Material**
(NCOMMS-22-30412).”

The authors thank the editor and reviewers, for their comprehensive and critical comments. The paper has been revised following the reviewers' comments. Details below. (*Black italic: Reviewer's comments; blue type: our response; red type: revisions; underline in figure descriptions: revised figure panels*)

Reviewer #1:

Comments: Though MDN usage as an additive in P3HT could be the first case from this manuscript, this strategy seems not to provide a breakthrough in terms of concept, economic prospect, and performance/stability.

Reply: Thanks the reviewer for the critical comments. We like to highlight the findings of this manuscript as following. Firstly, the introduction of the molecular bridge (MDN) has led to a dramatic increase in the efficiency of PSCs based on P3HT HTM, from 12.48% to 22.87%. Secondly, the stability has also been enhanced, that the PSCs based on M-P3HT has maintained 92% of its initial efficiency even after two months of aging at 75% relative humidity (RH) followed by one month of aging at 85% RH in the atmosphere. Moreover, their PCE has barely changed after operating at MPP under 1 sun illumination (~45 °C in N₂) over 500 hours, much better than the device based on P3HT (as shown in **Fig. 2f** and **Fig. 5f**). This is partially due to the superior light and humidity stability of P3HT as an HTM [Nature, 2019, 567, 511–515; Energy Environ. Sci., 2021, 14, 2419–2428; Angew. Chem. Int. Ed. 2021, 60, 16388–16393]. Therefore, in comparison of the widely adopted Spiro-OMeTAD HTM which is expensive, unstable, and requires the necessary dopants, MDN-P3HT, with low-cost and stable properties shows great advantages in not only performance and stability but also the accelerating progress of the industrialization and commercialization of PSCs.

Q1. A sentence “P3HT adopts the “edge-on” stacking ~ the non-radiative recombination loss of PSCs.” could deliver inappropriate meaning. Although ‘edge-on’ orientation is derived from direct contact between alkyl side chain and perovskite film, the evidence is still scarce to demonstrate how the orientation correlates to the junction properties.

Reply: Thanks for this constructive suggestion. As we referred to in the main text, reference 26, [Nature, 2019, 567, 511–515], “Despite the potential advantages of P3HT as an organic hole-

transport material (HTM) in perovskite solar cells, the resulting devices have a low open-circuit voltage (V_{oc}) due to additional non-radiative recombination at the perovskite/P3HT interface". And this has also been reported in several papers [Energy Environ. Sci., 2019, 12, 2778-2788; Adv. Mater., 2019, 31, 1902762], proving that in state-of-the-art solar cells, the non-radiative recombination at the interfaces between perovskite and P3HT is an important factor to determine the device performance. We believe that in our manuscript, our discussion "Nevertheless, P3HT adopts the "edge-on" stacking arrangement, that is the alkyl side chains directly contact perovskite film, resulting in an electronically poor contact at perovskite/P3HT interface, which can aggravate the non-radiative recombination loss of PSCs²⁶" is appropriate.

Q2. *Figure 1 (GIWAXS & SCLC) shows almost identical crystalline properties of P3HT layers regardless of molecular additives and the estimated hole mobilities are in the marginal range as well. So, further description must be required to help to understand why the quite similar results are important for this work. If not, a part related to Figure 1 might make confusion for readers.*

Reply: Thanks for your reminding. It is true that the crystalline properties and the hole mobility of P3HT, M-P3HT, R-P3HT have shown little difference. However, it is important to show these results, because these results have confirmed that the addition of MDN or RDN has not changed the molecular packing of P3HT. And this is different from the way that changing the "edge-on" packing of P3HT into "face-on" packing via SMe-TATPyr doping engineering in the previous work [Angew. Chem. Int. Ed. 2021, 60, 16388–16393]. After ruling out this possibility, we hence could further introduce the molecular bridge concept.

Q3. *It seems that the XPS result of Pb element and the description are opposite as compared to a precedent (<https://doi.org/10.1038/s41467-019-13909-5>). The previous paper suggested a malononitrile group in 6TIC-4F leads to lower binding energy of Pb element after being coated on a perovskite surface.*

Reply: Thanks for the kind reminding. After carefully checking the above paper, it is true that we have presented the opposite trend. However, we also noted that in that paper. The authors have conducted the calculation and claimed that "Therefore, after excluding the S-Pb bonding motif, the **N-Pb bonding motif** with a formation energy of -3.14eV is going to have an over 99% Boltzmann distribution ratio, which is likely to be the most effective **electron-donating group** to passivate most of the surface traps"(Fig. R1a).

While in our case, we have calculated the charge density difference by DFT and concluded that "It can be seen that the **electron-withdrawing groups of MDN/RDN** attract the electron from Pb ion in perovskite, while the degree of charge transfer in MDN- dimer complexes is greater than that in

RDN-one." (Fig. R1b). Hence, the difference of the electron deviation direction could be the reason of the observed difference in the XPS results.

Fig. R1. (a) Illustration of possible passivation mechanism and potential interaction sites (Ref: DOI: 10.1038/s41467-019-13909-5). (b) Charge density differences of PVK-V_i/MDN (this work).

Q4. A sentence "it is clear that as interfacial layer, ~ as shown in Fig. S18b." should be revised because GIWAXS pattern of M-P3HT does not demonstrate more ordered stacking as compared to that of pristine P3HT.

Reply: Sorry for this misunderstanding. In Fig. S18a, we tried to break the molecular bridge by introducing PS at M-P3HT/perovskite interface, that is to block electronic contact between MDN and perovskite. And in Fig. S18b we used MDN as an interface instead of dopant in P3HT, in order to weaken the π - π stacking between MDN and P3HT, that is to cut off the molecular bridge at MDN/P3HT side. Thereby, these two schematic figures are not for the comparison between P3HT and M-P3HT.

Reviewer #2:

Q1. It is clear that the stability of the PSCs based on M-P3HT HTM is enhanced. However, the insertion of MDN is more like doping in P3HT at a specific location, ca. at the perovskite/P3HT interface, the morphology stability of M-P3HT will be an important cause to the overall degradation of the PSCs. The authors should give a more detailed investigation about the degradation process of PSCs based on M-P3HT, especially from the respect of its morphology stability.

Reply: Thanks for this constructive comment. The SEM images of the fresh fabricated HTM/perovskite are illustrated in **Fig. S9**, where P3HT, M-P3HT, and R-P3HT uniformly covered the perovskite layer, and the outline of the perovskite crystals can be clearly observed due to their small thickness (~30 nm). Simultaneously, we have conducted the SEM characterization on the morphology of M-P3HT in PSCs after aging for three months in the same condition in **Fig. 5f**. From **Fig. S28**, on the surface of P3HT a large number of holes appear, which is possibly due to its weak protection of perovskite films leading to the degradation. By comparison, the surface of M-P3HT is still continuous and homogeneous, which means that M-P3HT could favor to inhibit the degradation of perovskite films.

Revision: “We then examined the surface morphology of the aged devices and found that both the P3HT and R-P3HT based devices showed varying degrees of porosity, particularly in the P3HT device, where the perovskite crystals had almost completely disintegrated (**Fig. S28**). For the M-P3HT device, however, the active layer of perovskite remains intact as crystals.”

Fig. S9. The top view SEM images with fresh devices of (a) perovskite, (b) perovskite/P3HT, (c) perovskite/M-P3HT, and (d) perovskite/R-P3HT.”

Fig. S28. The top view SEM images with aging devices of (a) perovskite/P3HT, (b) perovskite/M-P3HT, and (c) perovskite/R-P3HT.”

Q2. As listed in the introduction, such as references 29-32, the authors have compared several solutions to improve the efficiency based on P3HT. Clearly, some work has obtained a good result. I would suggest the authors to demonstrate the difference in order to highlight the novelty of the MDN molecular bridge.

Reply: Thanks for this constructive comment. We have added the difference in the revised manuscript.

Revision:” Clearly, researchers have taken various strategies, passivation of perovskite defects or modification of the P3HT hole mobility, to ameliorate the performance of P3HT in PSCs, whereas the poor contact issue at the perovskite/P3HT interface still has not been stressed.”

Q3. The authors have also designed a series of D-A and π -A molecules to prove that the D-D-A structure of MDN is an important reason of the enhanced device performance. Nevertheless, the design clue of the MDN is not clear. To make this work more scientific, I would suggest the authors give a clear demonstration about the molecular structure.

Reply: Thanks for this suggestive comment. In fact, MDN has been raised after a serious exploration of the molecular structure. Firstly, according to the literature, malononitrile can be a useful group to passivate perovskite film. Then we tried to p-p stack the molecular bridge with P3HT by thiophene. Whereas after the design and application of TDN into PSCs, it showed a higher crystallinity and poor solubility. Further, phenazine with alkoxy chains was adopted with the consideration of their higher solubility, and lower crystallinity. At last, on account of the matched

energy levels and hole mobility, a triphenylamine functional group was introduced, resulting in the MDN molecule.

Q4. In the PV performances part, it is interesting to notice that the hysteresis of the PSCs based on M-P3HT has been significantly mitigated. As far as I know, ion migration process could be an important reason, however, it is obvious that the adopted perovskite is the same. Therefore, what is the origin of this big difference between PSCs with P3HT, M-P3HT, and R-P3HT. The authors should give an explanation.

Reply: Thanks for this constructive suggestion. Ion migration in perovskite is an important cause of hysteresis, but it is not the only reason. It has been shown that the balance of carriers' transport in device, and the rapid release of photogenerated carriers at various interfaces, also have great influence on the hysteresis. (Adv. Energy Mater. 2017, 1700414; Chem.Rec., 2022, 22, e202100150; Energy Environ. Sci., 2018,11, 2404-2413). In our work, the defects at perovskite surface and the poor contact of P3HT/perovskite can cause non-radiative recombination, leading to a severe J–V hysteresis. For R-P3HT, the characterization in **Fig. 5** shows that RDN passivated the defects at perovskite surface, which can reduce the interfacial carrier recombination, leading to the reduced hysteresis. While for MDN, it forms a molecular bridge between perovskite and P3HT, thus mitigating the carrier accumulation at perovskite/HTM interface. At the same time, MDN has also been confirmed that it has passivated the defects at perovskite surface. We have also added the discussion in the revised manuscript.

Revision: “This can be explained with the constructed molecular bridge for efficient carrier transportation as well as the defect passivation³³⁻³⁵”

Q5. Despite the authors have shown a good result based on the MDN doped P3HT, the physical characterization of MDN is limited. Especially, the stability of MDN is of great importance, that its decomposition could possibly cause the degradation of perovskite films. Thus, the stability of the pure MDN or RDN should also been investigated, at least, the analysis of TGA to illustrate the decomposition temperature, should be conducted.

Reply: Thanks for this constructive suggestion. We have carried out TGA measurements on pure MDN and RDN as shown in **Fig. R2**. Both of the two compounds decompose at temperature beyond 380 °C. We, therefore, consider both MDN and RDN are stable.

Fig. R2. The TGA of MDN and RDN.

Q6. For all the NMR spectra, the reference is all missing.

Reply: Thanks for this kind remind. We have added the tetramethyl silane (TMS) reference in the supporting information.

Q7. In the supporting information, the method of CV characterization, "Ferrocene/ferrocenium (Fc/Fc+) was used as the reference", here I think the Ferrocene/ferrocenium (Fc/Fc+) was acting as a calibration or a reference electrode. Please clarify.

Reply: Thanks for this reminding. We have clarified it in the supporting information.

Revision: " Ferrocene/ferrocenium (Fc/Fc+) was used as the calibration."

Q8. Figure S11, the caption "J-V characteristics of the best-performing." Please complete the sentence.

Reply: Thank you for your kind remind. We have completed the sentence in the supporting information.

Revision:

Fig. S11. J - V characteristics of the best-performing based on the different HTMs (P3HT, 2-M-P3HT, 4-M-P3HT, 6-M-P3HT, and R-P3HT, respectively)."

Reviewer #3:

Q1. The authors need to state more discussion on the kinetics of the hole extraction at the interface of the perovskite/P3HT by carrying out TRPL or TA studies.

Reply: Thanks for the constructive suggestion. We have conducted the TRPL to discuss the charge transfer and recombination kinetics at the HTMs/perovskite interface. In **Fig. 3d**, and the average photoluminescence lifetime of perovskite, perovskite with P3HT, R-P3HT, and M-P3HT HTMs were determined to be 1071.71, 464.75, 397.38, and 302.377 ns, respectively (**Table S4**), indicating the faster carrier transfer from perovskite to HTM when MDN was adopted. In particular, the shorter photoluminescence lifetime (τ_1) correlates to the non-radiative recombination [Adv. Energy Mater.2020, 10, 1904134; Adv. Energy Mater. 2021, 11, 2003489; Nature Energy, 2022, 7, 708–717], suggesting the mitigated non-radiative recombination at perovskite/M-P3HT interface. We have also added the corresponding discussion in the revised manuscript.

Revision: "We have performed the steady-state photoluminescence (PL) and time-resolved PL (TRPL) to investigate the charge transfer kinetics at perovskite/HTM interface. To avoid the light absorption by P3HT and consist with the J - V characterization, the illumination was applied from the perovskite side. **Fig. 3c** shows the PL spectra, where different from the low intensity of device

“Due to their charge extraction effect, M-P3HT exhibits a strong PL intensity along with a significant blue-shift, from 799 to 792 nm, which indicates accompanied with a charge extraction process, the trap state density at the perovskite/M-P3HT interface has also decreased^{40,41}. In **Fig. 3d**, and the average photoluminescence lifetime of perovskite, perovskite with P3HT, R-P3HT, and M-P3HT HTMs were determined to be 1071.71, 464.75, 397.38, and 302.377 ns, respectively (**Table S4**), indicating the faster carrier transfer from perovskite to HTM when MDN was adopted. In particular, the shorter photoluminescence lifetime (τ_1) correlates to the non-radiative recombination⁴¹⁻⁴³, suggesting the mitigated non-radiative recombination at perovskite/M-P3HT interface, consistent with the steady-state PL results. This is direct evidence of the formation of the molecular bridge by introducing the MDN molecule.”

Fig. 3. Molecular bridge characterization of M-P3HT. High-resolution XPS spectra of (a) Pb 4f spectra, and (b) N 1s spectra. (c) Steady-state PL spectra, and (d) TRPL spectra”

Q2. In Figure S12 and Figure 3, it is found that MDN is more effective in passivating the perovskite surface than RDN. For clear comparison, the authors need to provide TCSPC measurement for two samples.

Reply: Thanks for the constructive comment. Despite we have tried to search the TCSPC equipment, we still could not conduct the characterization since the equipment we found is still under construction. Instead, in our work, we have adopted the SCLC (**Fig. 5a**), EIS (**Fig. 5b**), capacitance–frequency plots (**Fig. 5c**), voltage dependence on light intensity (**Fig. 5d**), and dark J – V curves (**Fig. 5e**) to characterize the trap state, and have obtained the conclusion that MDN is more effective in passivating the perovskite surface. Furthermore, as discussed in Q1 of reviewer 3, we have also conducted the PL and TRPL characterization to verify these results. We have also added the corresponding discussion in the revised manuscript.

Revision: “We have performed the steady-state photoluminescence (PL) and time-resolved PL (TRPL) to investigate the charge transfer kinetics at perovskite/HTM interface. To avoid the light absorption by P3HT and consist with the J-V characterization, the illumination was applied from the perovskite side. **Fig. 3c** shows the PL spectra, where different from the low intensity of device with P3HT and R-P3HT due to their charge extraction effect, M-P3HT exhibits a strong PL intensity along with a significant blue-shift, from 799 to 792 nm, which indicates accompanied with a charge extraction process, the trap state density at the perovskite/M-P3HT interface has also decreased^{40,41}. In **Fig. 3d**, and the average photoluminescence lifetime of perovskite, perovskite with P3HT, R-P3HT, and M-P3HT HTMs were determined to be 1071.71, 464.75, 397.38, and 302.377 ns, respectively (**Table S4**), indicating the faster carrier transfer from perovskite to HTM when MDN was adopted. In particular, the shorter photoluminescence lifetime (τ_1) correlates to the non-radiative recombination⁴¹⁻⁴³, suggesting the mitigated non-radiative recombination at perovskite/M-P3HT interface, consistent with the steady-state PL results. This is direct evidence of the formation of the molecular bridge by introducing the MDN molecule.”

Fig. 3. Molecular bridge characterization of M-P3HT. High-resolution XPS spectra of (a) Pb 4f spectra, and (b) N 1s spectra. (c) Steady-state PL spectra, and (d) TRPL spectra.”

Q3. In page 10, “It is clear that as interfacial layer, MDN fails to form a charge transportation channel, which is possibly due to its disordered stacking with P3HT, as shown in Fig. S18b.” is mentioned. Did the authors compare the ordering of P3HT on MDN/perovskite with the ordering of M-P3HT on the perovskite film?

Reply: Thanks for this constructive question. We have conducted the GIWAXS based on P3HT/MDN/perovskite as shown in Fig. S18, where the crystalline P3HT signal is weak. Thus, comparing with the GIWAX of M-P3HT/perovskite in Fig. S3b, we consider the ordering of P3HT on MDN/perovskite is worse than that of M-P3HT on the perovskite film.

Revision: “We have also conducted the GIWAXS based on P3HT/MDN/perovskite as shown in Fig. S18, where the crystalline P3HT signal is weak when comparing with that for M-P3HT/perovskite in Fig. S3b. Therefore, as an interfacial layer, MDN fails to form a charge transportation channel, as shown in Fig. S19b.”

Fig. S18. 2D GIWAXS patterns based on the structures of (a) Si-wafer/MDN, (b) Si-wafer/perovskite/MDN, and (c) Si-wafer/perovskite/MDN/P3HT.”

Fig. S19. Schematic diagram of the molecular stacking with (a) PS interface and (b) MDN interface.”

Q4. It is hard for the readers to identify the labelling and colors of MDN/P3HT complex in Figure 4e and 4f. It would be helpful if extended picture with a higher resolution is larger. In addition, top view SEM images of Figure S9 with higher resolution need to be given.

Reply: Thanks for this kind remind. We have updated Figure 4e, f and SEM in Figure S9 in the revised manuscript and supporting information.

Revision:

Fig. 4. Snapshots of the molecular bridge. Charge density differences of (a) PVK/MDN, (b) PVK- V_I /MDN, (c) PVK/RDN, and (d) PVK- V_I /RDN. PVK- V_I denotes the perovskite with I vacancy, and ΔE expresses the binding energy. Local perovskite is displayed for clarity. The yellow and cyan regions represent electron accumulation and depletion, respectively. Independent gradient model based on Hirshfeld partition (IGMH) of MDN/P3HT dimer complexes with different geometries (e) and (f). ”

Fig. S9. The top view SEM images with fresh devices of (a) perovskite, (b) perovskite/P3HT, (c) perovskite/M-P3HT, and (d) perovskite/R-P3HT. ”

Q5. In Figure 2a and Figure S8, the thickness of P3HT is found to be ~ 30 nm. It is also hard to distinguish P3HT layer from three layers of perovskite, P3HT and Ag. Is a thin (~ 30 nm) layer of P3HT enough to cover the rough surface of the perovskite film? The authors need to make this clearer.

Reply: Thanks for this suggestive comment. From the high-resolution SEM image in **Fig. S9**, in spite of the small thickness of the HTM layer, it can still form a continuous and homogeneous film. This is also the case for organic solar cells, where the active layer in OPV is also as thin as 30-40 nm.

Q6. In Figure 5e, dark current densities of P3HT and R-P3HT devices are much higher than that of M-P3HT device. What is the main reason for this difference? Is such a trend consistent with SCLC results from Figure S27?

Reply: Thanks for this constructive question. Several studies (InfoMat, 2020, 2, 1247–1256; Sci. Adv. 2021; 7: eabg6716; Adv. Energy Mater., 2018, 9, 1803135) have shown that dark currents are mainly generated by leakage currents from trapped carriers or carriers injected under reverse bias. They are closely related to defects in perovskite, hence interfacial engineering, increasing perovskite crystal size, and defect passivation can significantly reduce dark currents. In our case, M-P3HT drastically reduces the surface defects of perovskite, leading to a much smaller dark current density comparing with those for devices based on P3HT and R-P3HT. Moreover, the presence of molecular bridge also leads to a significant reduction in carrier accumulation at the perovskite/P3HT interface, favoring the decrease of the leakage effect. In **Fig. S27**, the hole mobility of perovskite/M-P3HT is one order of magnitude larger than that of perovskite/R-P3HT, two orders of magnitude larger than that of perovskite/ P3HT. The trend is consistent.

REVIEWER COMMENTS

Reviewer #1 (Remarks to the Author):

Despite the author's effort, the questions raised before still are unsolved and the explanation couldn't touch a basic intention to make this manuscript clear and valuable.

It is well-known fact that P3HT has much potential for commercial perovskite solar cells as hole transport material. However, it might not be guaranteed that the molecular bridge suggested by the authors made a breakthrough. Previous papers have already claimed the strategy to introduce alkyl ammonium derivatives between P3HT and perovskite to improve performance and stability. Although alkyl ammonium derivatives are cheap and easy to purchase large scale, MDN should have a higher cost due to several synthetic steps. Moreover, the figure of merits from MDN concept didn't show advanced performance and stability as compared to previous reports. So, it is difficult to estimate whether the MDN strategy provides a breakthrough enabling to be published in Nature Communications.

Especially, please consider a correlation between orientation of P3HT and interface properties again. Poor interface contact must lead to non-radiative recombination. However, it is under dispute how the orientation of P3HT influences on recombination behavior and performance. Although some researchers suggested "edge-on" orientation could result in more efficient charge transfer [J. Am. Chem. Soc. 139, 3378 (2017)], there is not well-established fundamental for the correlation. So, authors should be prudent to discuss the issue.

Reviewer #2 (Remarks to the Author):

revision is OK.

Reviewer #3 (Remarks to the Author):

All the issues raised by the reviewers are well-addressed. Thus, I would recommend the publication of this work in Nature Comm.

- As for Figure 3d and Table S4, the authors need to confirm the deconvolution fitting from the result of TRPL decay (i.e. A1 for perovskite sample is 2.09)

Response to the reviewers' comments for the paper entitled
**“Constructing Molecular Bridge for High-Efficiency and Stable Perovskite Solar Cells
based on P3HT Hole Transport Material
(NCOMMS-22-30412A).”**

The authors thank the reviewers, for their comprehensive and critical comments. The paper has been revised following the reviewers' comments. Details below. (*Black italic: Reviewer's comments; blue type: our response; red type: revisions; underline in figure descriptions: revised figure panels*)

Reviewer #1:

Q1: It is well-known fact that P3HT has much potential for commercial perovskite solar cells as hole transport material. However, it might not be guaranteed that the molecular bridge suggested by the authors made a breakthrough. Previous papers have already claimed the strategy to introduce alkyl ammonium derivatives between P3HT and perovskite to improve performance and stability. Although alkyl ammonium derivatives are cheap and easy to purchase large scale, MDN should have a higher cost due to several synthetic steps. Moreover, the figure of merits from MDN concept didn't show advanced performance and stability as compared to previous reports. So, it is difficult to estimate whether the MDN strategy provides a breakthrough enabling to be published in Nature Communications.

Reply: Thanks for this critical comment. However, we consider there must be some misunderstandings here.

Firstly, the work previously reported in the literatures are different from our work. As referred in [Nature, 2019, 567, 511–515], the authors introduced an alkyl ammonium derivative (HTAB) to synthesize a thin layer of wide-bandgap halide perovskite to enable a relatively weak interaction between aliphatic chains in DHA and P3HT. This could promote the self-assembly of P3HT nanofibrils, thus resulting in higher hole mobility. However, this is different from our work. In our work, with various characterizations, we have proved that the packing of P3HT has not been changed with MDN. Instead, we build a molecular bridge, MDN, between perovskite and P3HT, to give charge a transporting channel. We have indeed cited and compared the relevant work with the results in our paper.

Secondly, about the cost, despite the molecular structure of MDN is relatively complex when compared with this n-hexyl trimethyl ammonium bromide (HTAB), the MDN usage quantity is quite low (4 mg/mL), and the HTAB still needs a further reaction. Therefore, we think it would be difficult, and also not reasonable, to compare our work in cost with [Nature, 2019, 567, 511–515], especially when two different methods are adopted.

Lastly, regarding to the performance and stability, we have also exhibited a three-months (>2100 h) stability of the unencapsulated devices, much longer than the reported 1,000 h. And it also showed relatively good light stability.

Q2: Especially, please consider a correlation between orientation of P3HT and interface properties again. Poor interface contact must lead to non-radiative recombination. However, it is under dispute how the orientation of P3HT influences on recombination behavior and performance. Although some researchers suggested “edge-on” orientation could result in more efficient charge transfer [J. Am. Chem. Soc. 139, 3378 (2017)], there is not well-established fundamental for the correlation. So, authors should be prudent to discuss the issue.

Reply: Thanks for this suggestive advice. We agree with that “authors should be prudent to discuss the issue”, and we have changed the corresponding discussion in the revised manuscript.

However, we consider that to further explore the correlation between orientation of P3HT and interface properties is already beyond our research scope. In our work, we found that the interfacial contact of perovskite and edged-on packed P3HT is poor, thus without changing the packing style of P3HT, we improved the charge transport properties. It is different from the commonly adopted method, that is to change the packing style of P3HT, as reported in the literatures [Nature, 2019, 567, 511–515; Angew. Chem. Int. Ed. 2021, 60, 16388-16393]. But, we agree that it would be another important and interesting topic for the future research.

Revision: “Nevertheless, P3HT adopts the “edge-on” stacking arrangement, that is the alkyl side chains directly contact perovskite film, which was found presenting an electronically poor contact at perovskite/P3HT interface, which can aggravate the non-radiative recombination loss of PSCs²⁶.”

Reviewer #2:

Comments: revision is OK.

Reviewer #3:

As for Figure 3d and Table S4, the authors need to confirm the deconvolution fitting from the result of TRPL decay (i.e. A1 for perovskite sample is 2.09).

Reply: Thank you for this kind remind. We have revised the manuscript accordingly.

Revision: “In **Fig. 3d**, the deconvolution fitting from the result of TRPL decay (**Table S4**), and the average photoluminescence lifetime of perovskite, perovskite with P3HT, R-P3HT, and M-P3HT HTMs were determined to be 1071.71, 464.75, 397.38, and 302.377 ns, respectively, indicating the faster carrier transfer from perovskite to HTM when MDN was adopted.”